# Analysing Researchers’ Engagement in Research Hospitals: A Pilot Study in IRCCS—Italian Research Hospitals

**DOI:** 10.3390/healthcare10122458

**Published:** 2022-12-05

**Authors:** Giulia Mollica, Rosario Caruso, Gianluca Conte, Federico Ambrogi, Sara Boveri

**Affiliations:** 1Grant Office, Scientific Directorate, IRCCS Policlinico San Donato, San Donato Milanese, 20097 Milan, Italy; 2Department of Biomedical Sciences for Health, University of Milan, 20100 Milan, Italy; 3Health Professions Research and Development Unit, IRCCS Policlinico San Donato (MI), San Donato Milanese, 20097 Milan, Italy; 4Department of Clinical Sciences and Community Health, University of Milan, 20100 Milan, Italy; 5Laboratory of Biostatistics and Data Management, Scientific Directorate, IRCCS Policlinico San Donato, San Donato Milanese, 200097 Milan, Italy

**Keywords:** surveys, qualitative methods, organisation science, research engagement, research hospitals

## Abstract

Despite universities in the UK, USA and Australia having developed tools and strategies to enhance academic engagement, there is a paucity of information from universities and research hospitals where conceptual and theoretical research still appears more heterogeneous. In Italy, there is a growing recognition that more needs to be done to define strategies to improve research engagement. Italian research hospitals are represented by the Scientific Institute for Research, Hospitalization and Healthcare (named IRCCS from the Italian acronym of these organisations), representing the best of Italian research and the National Health Service System. This article provided a pilot description of research engagement in a representative Italian IRCCS hospital for the first time. A pilot study was developed, and a brief questionnaire was validated to explore research engagement. The identified clusters of researchers’ engagement were provided to describe an initial theory-grounded framework. Based on the perspective of research administrators and the Ministry of Health during round tables, the developed questionnaire identified two clusters of researchers’ engagement and measured “Project-oriented” and “Organisation-oriented” engagement. Among the responders, IRCCS senior researchers tended to have higher project-oriented engagement, while young researchers had slightly higher organisation-oriented engagement. The contribution of this article is a hypothesised two-level theory-grounded framework to study and improve research engagement activities and strategies in a research hospital, with the potential for an overlap with other European research institutions.

## 1. Introduction

Throughout history, the missions of universities have evolved and been influenced by different national contexts. High technology and rapid globalisation have altered work, leisure time, and formal schooling structures. Yet, they must remain flexible enough to respond to emerging social demands, technological change, and economic realignments [1]. Generating knowledge through teaching and research activities remains one of the key missions of organisations such as universities. The evolution of the university over the years led to the introduction of another main mission, called the “third mission”, as a re-thinking, re-distribution and re-contribution of knowledge to social and economic progress [2,3]. In this context, university and research hospitals have been influenced by different national contexts through the years; they have contributed to the education system, generating knowledge and facilitating technology transfer. The role of hospitals in developing, applying, testing, and promoting medical knowledge and innovation has been discussed greatly [4,5,6,7], and past studies have considered hospitals, in comparison with universities, much weaker in the innovation role [8].

In Italy, research hospitals actively involved in training, innovative research activities, and producing new medical knowledge are represented by the Scientific Institute for Research, Hospitalization and Healthcare (named IRCCS from the Italian acronym of these organisations) [9]. IRCCS hospitals represent the best of the Italian research and National Health Service Systems (NHSS), with high standards of healthcare research and staff training. The Italian NHSS has been characterised by a pattern of innovation and knowledge, followed by cost-containment measures and re-organisation. NHSS created 20 healthcare systems, exacerbating the differences in access to care between the North and South to delicately balance centralised versus regional and local control [10]. Therefore, IRCCS hospitals are the institutes involved in the Italian medical healthcare innovation process that NHSS make available to patients, and clinical research allows them to develop innovation as leading organizations or as partners with other organisations in the medical–industrial complex and technology transfer. Currently, 52 Italian hospitals have obtained recognition as IRCCS hospitals, of which 22 are public and 30 are private [9]. The Ministry of Health provides yearly economic support to all the IRCCS hospitals [11] and promotes results that are directly transferable to the NHSS and help it achieve its strategic objectives.

In 2006, IRCCS Policlinico San Donato (IRCCS PSD), which is based in Milan, was recognized by the Italian Ministry of Health as a Scientific Institute for Research, Hospitalisation and Healthcare to study and treat cardiac and great vessel diseases. Since 2009, it has also been the educational site of the faculties of Medicine and Surgery, Nursing Sciences, and different medical specialization schools of the University of Milan, and a training centre for numerous foreign physicians. Since 2020, IRCCS PSD has been the coordinator of the Italian Cardiovascular IRCCS Network, promoted by the Ministry of Health and established in 2017.

The Italian Ministry of Health defined young Italian IRCCS researchers (YRs) as researchers aged under 40 years and senior IRCCS researchers (SRs) as researchers aged over 40 years. YRs and SRs are the basis for building innovation, competitiveness, and excellence for the country. Increasing the research engagement of YRs and SRs is pivotal for both the Italian Government, the IRCCS hospitals network, and each IRCCS research institution’s performance due to the positive associations between research engagement, scientific productivity, research quality, and positive research culture, and might enhance the organisational capacity of IRCCS hospitals in producing high-quality research partnering with research end-users. Furthermore, IRCCS hospitals collaborate by design with universities to train future medical practitioners and researchers.

Academic engagement represents the interaction between researchers and research end-users and/or research partners (outside and inside academia or research organisations) for the mutually beneficial transfer of knowledge, technologies, methods, or resources [12]. The importance of engagement as a critical process of change can also be seen as part of a long tradition within social science research, and was described by Glerup and Horst [13] as integration rationality which conceptualizes knowledge production as a fundamentally collaborative process. In 2013, Perkmann et al. clarified the concept of academic engagement, stating that academic engagement is represented by inter-organisational collaboration instances, usually involving “person-to-person interactions” that link universities and other organisations for a fee or non-financial benefits [2]. Academic engagement is a high-level government priority and a fundamental part of research methodologies and institutional mandates in many countries, such as the United Kingdom, Germany, United States of America, Canada, and Australia [14,15].

Although some countries have developed tools, expertise, and strategies to enhance the academic engagement of researchers in hospitals using theory-informed and evidence-based paradigms [16,17,18,19,20], in Italian IRCCSs, in the national context, decision-makers at the Ministry of Health, research managers and administrators, and scientists have paid little attention to define strategies to increase research engagement.

Given the lack of available information on Italian research engagement in IRCCSs, we proposed this pilot study to identify a framework to guide research exploring strategies to enhance research engagement before implementing pan-national initiatives in the cardiovascular IRCCS hospital network in all the Italian IRCCSs and in similar European organisations. The pilot study aimed (1) to develop and validate a brief questionnaire to explore research engagement in IRCCS YRs and SRs; (2) identify the clusters of the research engagement’s levels; (3) and propose a theory-grounded framework to guide future research endeavours in the field of Italian research engagement in research hospitals.

## 2. Materials and Methods

This article presents preliminary findings from a pilot study in one of the most representative Italian IRCCS hospitals, which developed an e-survey exploring, for the first time, strategies to enhance research engagement in the context of Italian hospitals that may overlap with university engagement environments; after all, both organisations have the same three key missions (teaching, research, knowledge sharing). Starting from Perkmann’s recent literature review about academic engagement [21] and Thune and Mina’s review [8] on the role of the hospitals as innovators, this article focuses on the particular need to improve research engagement activities and strategies in this complex environment, potentially overlapping other European institutions (university hospitals or research hospitals). Based on the point-of-view of research managers and administrators, and the Ministry of Health, the e-survey aimed to describe and to classify IRCCS researchers with self-report questionnaires.

### 2.1. Design

This study utilised a three-phase design (Figure 1), consistent with the questionnaire design recommendations to develop self-report questionnaires [22]. Phase one aimed at conceptualising and generating the pool of items to measure research engagement, exploring the views of Italian IRCCSs’ research administrators during round tables organized by the Ministry of Health between December 2017 and February 2019. The aim of the round tables was to promote a concrete modification to the national guidelines to increase the visibility of IRCCS hospitals in Europe.

Phase two was based on a pilot, mono-centre, cross-sectional data collection through an e-survey performed at IRCCS PSD from June 2019 to July 2019. A checklist for reporting results of the internet e-survey (CHERRIES) was used to ensure a complete description of the adopted e-survey of phase two (see Appendix A) [23]. An anonymous and voluntary e-survey was created using SurveyMonkey™ and there was a question to assess whether the respondent was a YR or SR researcher. The e-survey was intended to test research engagement through ten questions, and no professional characteristics were collected to describe the sample because the main goal of collecting the data was to validate the questionnaire with a scoring procedure rather than portraying an epidemiological description of research engagement.

Phase three encompassed a consensus discussion that aimed to generate an initial theory-grounded framework, with consideration given to the current literature, to guide future research endeavours in the field of research engagement in Italy, with a possible future scaling-up to similar European institutions. The framework was represented in graphical form, which was critically discussed and modified until a consensus was obtained.

### 2.2. Scoring Procedure and Statistical Analysis

Based on the content of each response, an a priori strategy was defined to transform the responses into a binary score for each question: engaged = 1; not engaged = 0 (see Appendix A).

A principal component analysis (PCA) was performed to support an evidence-based scoring procedure, also providing an initial validation of the adopted questionnaire (aim 1). PCA presented the pattern of similarity in responses to the questionnaire by showing similarities as principal components derived from the linear combinations of the original variables.

PCA scores were used as data for a cluster analysis that visualized the association among psychological tests, and to plot variables and subjects together using a biplot.

A scree plot was used to determine the number of meaningful principal components to retain for cluster analysis. After determining the number of principal components, an oblique rotation of the Kaiser-normalised matrices (Promax) was employed to facilitate the understanding of the relationships between responses transformed into binary variables (engaged vs. unengaged) and principal components. Once the principal components were interpreted, the scoring procedure was based on calculations of the percentage of engaged answers among the questions connected with each principal component.

A cluster analysis was performed using individual responses to place groups (clusters) with similar answers (aim 2). The clustering was performed by adopting Ward’s hierarchical method, considering a range of possible solutions based on the error sum of squares. The most suitable solution was given by the model that encompassed more stable and interpretable clusters.

The number of clusters (k) to be considered was identified using the following 3 indices: cubic clustering criterion (CCC), pseudo F (PSF), and T2 (PST2).

The hierarchical structure of the data was visualized using a dendrogram.

Descriptive statistics were used to describe differences in the answers between each cluster and between YRs and SRs. Then, the computed scores of engagement were described for each cluster and compared using a Mann–Whitney U test. All statistical tests were two-tailed, and *p* < 0.05 was considered statistically significant. Analysis was performed using SAS software, version 9.4 (SAS Institute, Inc., Cary, NC, USA), and R software, version 4.0.5.

## 3. Results

The presentation of the findings of this study begin with an overview of how IRCCS YRs and SRs answered after an a priori strategy to encode survey questions in engaged and not-engaged researchers.

### 3.1. Principal Determinants Described by PCA and Cluster Analysis

Seventy-seven researchers were invited to participate in this pilot study (30 (39.0%) SRs, and 47 (61.0%) YRs). The survey was compiled by 50 researchers (response rate = 65%), of which there were 30 SRs (response rate = 100%), and 20 YRs (response rate = 43%).

The PCA model (total variance = 33.59%) identified two principal components as the most plausible dimensionality of the questionnaire. The questions (Table 1) best represented by principal component 1 that showed a predicted variance equal to or greater than 10% were questions 2, 3, 5, and 7. The questions best represented by principal component 2 that showed a predicted variance equal to or greater than 10% were questions 1, 4, 8, 9, and 10. Question 6 was not significantly predicted by either principal component 1 or principal component 2. Principal component 1 was labelled as “Project-oriented engagement”, considering the underlying meaning of questions 2, 3, 5, and 7. Principal component 2 was labelled as “Organisation-oriented engagement”, considering the underlying meaning of questions 1, 4, 8, 9, and 10. The mean overall scores of “Project-oriented engagement” and “Organisation-oriented engagement” were 77% (standard deviation, SD = 15) and 81% (SD = 15), respectively.

The researchers were divided into two clusters based on the similarity of responses within each possible cluster (Figure 2).

Table 1 shows questions in cluster 1 (*n* = 29) and cluster 2 (*n* = 13). The majority of SRs (*n* = 14; 93.3%) were in cluster 1, with 15 YRs (55.5%). Researchers answering that they were aware of a figure with the role of project reviewer inside the organisation (question 4) were more frequent in cluster 1 than in cluster 2 (89.7% vs. 46.2%; *p* = 0.005). It is likely that, in cluster 1, more researchers were involved in meetings and courses for project preparation (*p* < 0.001).

Figure 3 shows the scores of “Project-oriented engagement” and “Organisation-oriented engagement” in the two clusters of responders, highlighting a non-statistically significant but slightly higher score of “Project-oriented engagement” in cluster 1 (*p* = 0.360), and a non-statistically significant but slightly higher score of “Organisation-oriented engagement” in cluster 2 (*p* = 0.420).

### 3.2. Future Implementations Based on Survey Determinants

The initial theory-grounded framework proposal derived from this pilot project is described in Figure 4. A two-level model was hypothesised, considering the theoretical interconnection between individual-level engagement and its consequences and organisational characteristics [24]. Project-oriented and organisation-oriented engagement might positively empower researchers on an individual level, and the interplay between engagement, mentoring, and empowerment might also positively impact performance. At the organisational level, the interplay between structural orientation to research engagement, the structural research ecosystem (availability of adequate professional roles and tech-equipped infrastructure), and management style (person-oriented, coaching) might influence individual and organisational performance levels.

## 4. Discussion

### 4.1. Statement of Principal Findings

The results showed that SRs appeared to be more project-oriented engaged, while approximately half of the enrolled YRs were project-oriented, and the other half of the sample was more organisation-oriented engaged. Although organisation-orientated engagement did not differ significantly between clusters, the cluster encompassing the majority of SRs (cluster 1) seemed to have, on a descriptive basis, a slightly lower focus on organisation-oriented engagement. This initial trend suggests the need for future investigating scenarios that in the current study were not feasible considering the pilot nature of the research. In addition, due to the different gender engagement paths previously described in the literature [25] and our recent national policies to support gender equality, future specific analysis and action ensuring equal access to engagement needs to be evaluated and discussed.

The collaborative round table, supported by the Ministry of Health and involving all the Italian IRCCS research administrators, was a key feature of the e-survey design. The developed questionnaire showed adequate evidence of initial validity (dimensionality) considering “Project-oriented engagement” and “Organisation-oriented engagement”. The dimensionality emerging from this pilot study might be useful to guide future in-depth developments for the measurement of research engagement using a self-report approach. In fact, the lack of shared and theory-grounded measurements for assessing research engagement undermines the possibility of international research [21]. The developed measurements for assessing research engagement in this pilot study are also pivotal for planning future studies aimed at describing which individual- or organization-level factors may influence research engagement. Describing the determinants of research engagement in a future scaling-up of this research would provide guidance for planning actions to improve research engagement at local and national levels.

Phase three of the design (the framework proposal) highlighted the perceived importance of providing elements that could be useful to design research protocols encompassing multi-level assessments. Research organisations are made of interacting layers (organisational and individual layers), and between layers there is, theoretically and empirically, some inter-dependence influencing the outcomes, such as performance [21]. In such a complex scenario, the concept of engagement of the individuals involved in organized settings is a promising lever for innovation [24].

In the proposed framework, the empowerment of researchers was included, considering research engagement as an antecedent of empowerment. This conceptualisation aligned with the traditional definition of Kanter’s theory of empowerment: the ability to complete tasks successfully within an organisation by having access to information, resources, opportunities, and support [26]. Given that research engagement is the purposeful interaction between researchers and research end-users and/or research partners [16,27], and considering that it might be “Project-oriented” and “Organisation-oriented”, it is theoretically plausible that higher levels of research engagement allow researchers to achieve more successful tasks within their organisations. Mentoring novice researchers might support the relationships from research engagement to empowerment, as mentoring activities might trigger the effects of being engaged to become empowered.

The implications of increasing the levels of research engagement may also be important in relation to the general rights of patients. In fact, in the era of eHealth data and digitalization, levels of research engagement may also be considered important for ensuring the highest compliance to several regulations aimed at protecting the rights of patients, as per the case of the e General Data Protection Regulation and the European Charter of Patients’ Rights [28]. In this regard, research engagement is needed to guarantee the required standards of safe access to health information.

### 4.2. Strengths and Limitations

Pilot studies typically provide information on the feasibility of investigating specific topics and preliminary descriptive information. The sample size was not powered to determine differences among sub-groups, so generalising the results has to be considered with caution and they should be considered preliminary information. Future research with a complete description of the characteristics of responders might reveal more in-depth insights in order to understand the patterns between the individual characteristics of researchers and their engagement. Additional socio-demographic and professional characteristics in future studies are key to allow researchers an in-depth evaluation of the associations between the characteristics that potentially influence research engagement and the self-reported research engagement. In other words, along with the focus on describing differences between SRs and YRs, some other characteristics have to be further investigated in the scaling-up of this project, such as sex, type of education of researchers, regions where they work, ethical competency, and so on. This additional information may play a role in further clustering the research engagement levels. Finally, enriching the questionnaire with additional items to investigate engagement toward technology transfer activities using a larger and higher-powered sample is pivotal to provide a more robust validation of the initially proposed measurement of research engagement.

### 4.3. Interpretation within the Context of the Wider Literature

Most of the individual and organisational determinants of academic engagement described by Perkmann in 2013 and 2021 [2,21] were deeply discussed and compared to the IRCCSs’ context. As individual characteristics play an important role in predicting academic engagement, we included questions about seniority, grant funding, and previous commercialisation experience. We did not include gender analysis due to the very recent pro-women policies adopted in our country that may have imbalanced the gender gap [25]. Due to the ambiguous effect of organisational determinants on academic engagement [21], and the peculiarity of the IRCCS environment, we decided to include questions about specific organisational support with the presence of formal technology transfer infrastructures. For instance, the question about technology transfer activities was not significantly predicted. This result might reveal the engagement toward technology transfer activities, but it could not be expressed in the current version of the questionnaire. Moreover, in 2021, peer orientation influenced this process [21], so we confirmed that peer effects were stronger for YRs than SRs.

### 4.4. Implications for Policy, Practice, and Research

Overall, research engagement might support organisational improvements, for example, by creating an orientation to engagement at the organisational level [29], supporting a managerial style (e.g., people-oriented styles and/or coaching) [30], and the need for a research infrastructure encompassing adequate professional roles (e.g., Research Administrators) and tech-equipped settings [31]. Strategic centralised support for research engagement enables a coordinated collection of information about activities and impact, while providing a valuable resource as a body of knowledge for management, marketing, capacity building, interdisciplinary connections, partnerships, and reports for governments and funders [16].

## 5. Conclusions

This pilot study has provided an exploratory description of research engagement in an Italian IRCCS hospital, developing a brief questionnaire to allow a preliminary description of the research engagement levels’ correlated variables at an individual and organisational level, and proposing an initial theory-grounded framework for guiding future research endeavours in the field of research engagement in Italy. This research paves the way for further investigations aimed at understanding which elements may influence research engagement by using the developed measurement of research engagement and, therefore, planning tailored policy and educational interventions to sustainable strategies for improving research engagement.

## Figures and Tables

**Figure 1 healthcare-10-02458-f001:**
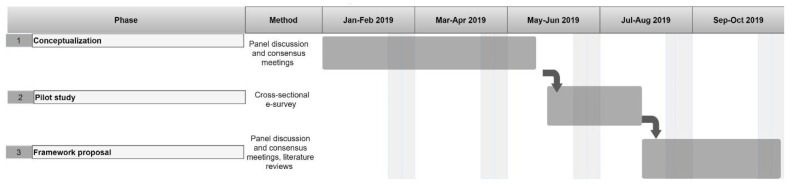
The three-phase design of the study, including the questionnaire design and recommendations to develop self-report questionnaires.

**Figure 2 healthcare-10-02458-f002:**
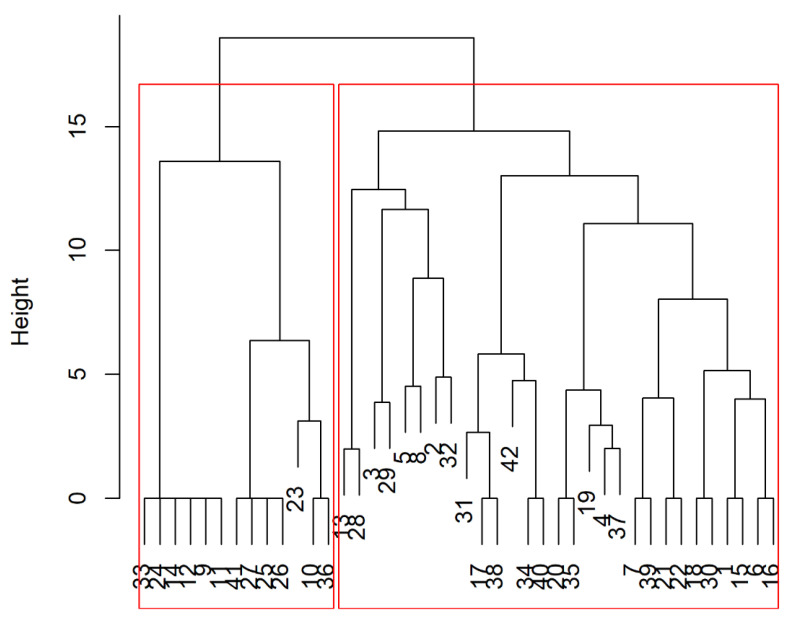
Hierarchical clustering tree based on survey answers and showing the similarity of responders in two different clusters.

**Figure 3 healthcare-10-02458-f003:**
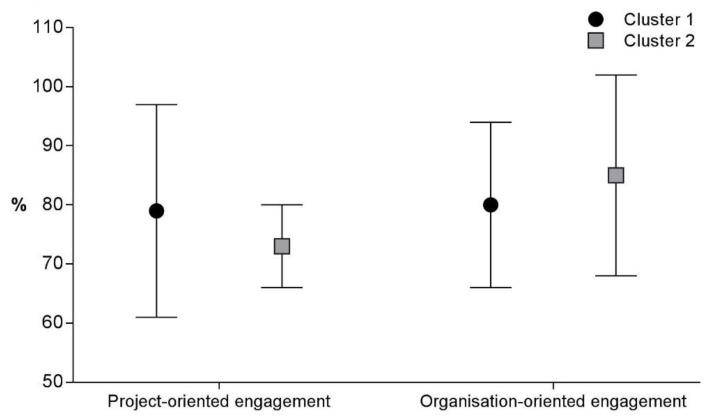
Mean and standard deviation of scores for “Project-oriented engagement” and “Organisation-oriented engagement” of two clusters.

**Figure 4 healthcare-10-02458-f004:**
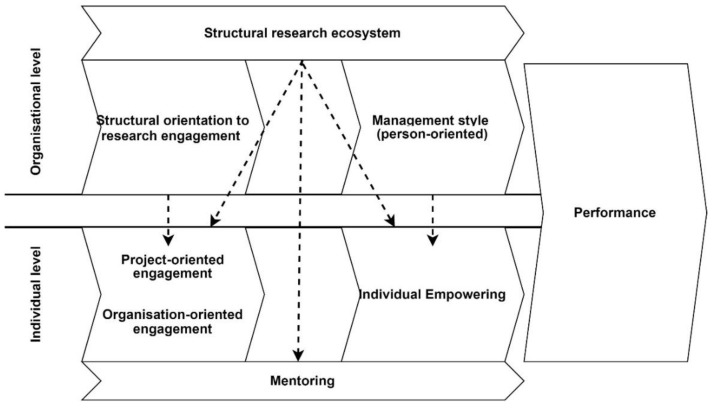
The proposed theory-grounded framework.

**Table 1 healthcare-10-02458-t001:** Description of answers indicating “Engagement” from the a priori codification in cluster 1 and cluster 2.

	Cluster 1 (*n* = 29)	Cluster 2 (*n* = 13)	*p*-Value
Have you ever applied for funding?	27 (93.1)	13 (100)	1.00
2. During the preparation of a project, on which aspect would you prefer to be mostly supported?	26 (89.7)	13 (100)	0.54
3. Would you find reading abstracts of funded projects, together with reviewers’ comments, as a comparison method?	27 (93.1)	13 (100)	1.00
4. In addition to your unit head, is there a figure inside the structure with the role of project reviewer in the pre-submission phase?	26 (89.7)	6 (46.2)	**0.005**
5. During project preparation, have you ever analysed the potential long-term impact of the projects through technology transfer activities?	20 (69.0)	12 (92.3)	0.13
6. During project preparation, who are your Institutional contact persons for assessing the potential impact of the project through technology transfer activities?	22 (75.9)	13 (100)	0.08
7. Have you ever participated in internal/external courses/meetings related to the preparation of project proposals?	19 (65.5)	0	**<0.0001**
8. Would you be interested in participating in internal/external courses/meetings to support researchers through online platforms or face-to-face courses?	14 (48.3)	10 (76.9)	0.10
9. Would you have the opportunity to organize face-to-face courses at your facility?	22 (75.9)	13 (100)	0.08
10. During the preparation of a funding request, would you need support for planning communication and dissemination activities?	27 (93.1)	13 (100)	1.00

## Data Availability

The data underlying this article will be shared on reasonable request to the corresponding author.

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
