# Peer review of "Analysing Researchers’ Engagement in Research Hospitals: A Pilot Study in IRCCS—Italian Research Hospitals"

_healthcare, 2022, doi:10.3390/healthcare10122458_

Round 1
Reviewer 1 Report
The authors took up a topic that is important for shaping health policy and takes into account public health. Since this is a pilot study, it means that it is an initial approach to identifying conditions for further research aimed at improving the efficiency of research centers. This is just a preliminary scout about the choice of research by SRs and YRs, but it is not enough.
It is a pity that at this stage the authors did not make a more detailed division of YRs and SRs, e.g. into women and men, type of education of researchers, regions where they work, ... .
However, the most important thing will be to identify the reasons that determine the choices of researchers. This is important even at the pilot study stage. In my opinion, the article lacks research questions for further analysis, which will give more knowledge about subgroups making similar choices, and above all, the reasons why they do so. Knowledge of these determinants, reasons and goals is of particular importance for health policy makers. Knowing these reasons, it will be possible to construct incentives to ensure that the decisions made by researchers will enable the achievement of research, health policy and public health goals to the greatest extent possible.
Reviewer 2 Report
Dear authors, thank you for the very interesting study. I would suggest adding a paragraph about implication to the patient rights, especially in the way of digitalization, an example is explained here https://ieeexplore.ieee.org/document/9941532 )
As well, it will be interesting to include ethical principles in the research, is there any problem for the doctor - clinicians, and researchers at the same time to get ethical approval for the clinical trials for example? It can affect also active engagement in the hospital.
please extend the conclusion with concrete data and future work,., recommendations...
Reviewer 3 Report
The main aims of this pilot study are: developing and validating a brief questionnaire to explore research engagement in IRCCS YRs and SRs; identifying the clusters of the research engagement's levels; proposing a theory-grounded framework to guide future research endeavours in the field of Italian research engagement in research hospitals.
The topic is original and relevant in the field. The authors could improve the discussion of the results.
The conclusions are consistent with the evidence and the arguments presented and the references are appropriate.
The table and the figures are clear and necessary
in the discussion explain the results, please.
Ex: why the SRs appeared to be more project-oriented engaged, while roughly half of the enrolled YRs were project-oriented, and the other half of the sample was more organisational-oriented engaged.
Reviewer 4 Report
My most sincere congratulations for the work done in this pilot work. I would like you to continue investigating this in order to confirm your results with future research. I really liked your work in general, the only thing I would like to point out would be the following:
Line 144 – Please, improve the quality of the image or modify it to make the letters look better.
